# Comparison of the Molecular Motility of Tubulin Dimeric Isoforms: Molecular Dynamics Simulations and Diffracted X-ray Tracking Study

**DOI:** 10.3390/ijms242015423

**Published:** 2023-10-21

**Authors:** Tsutomu Yamane, Takahiro Nakayama, Toru Ekimoto, Masao Inoue, Keigo Ikezaki, Hiroshi Sekiguchi, Masahiro Kuramochi, Yasuo Terao, Ken Judai, Minoru Saito, Mitsunori Ikeguchi, Yuji C. Sasaki

**Affiliations:** 1Graduate School of Medical Life Science, Yokohama City University, 1-7-29 Suehiro-cho, Tsurumi-ku, Yokohama 230-0045, Japan; ekimoto@yokohama-cu.ac.jp (T.E.); minoue@yokohama-cu.ac.jp (M.I.); ike@yokohama-cu.ac.jp (M.I.); 2HPC- and AI-Driven Drug Development Platform Division, Riken Center for Computational Science, RIKEN, 1-7-22 Suehiro-cho, Tsurumi-ku, Yokohama 230-0045, Japan; 3Department of Medical Physiology, Kyorin University School of Medicine, 6-20-2 Shinkawa, Mitaka 181-8611, Japan; nakayama@ks.kyorin-u.ac.jp (T.N.); yterao@ks.kyorin-u.ac.jp (Y.T.); 4Graduate School of Frontier Sciences, The University of Tokyo, 5-1-5 Kashiwanoha, Kashiwa 277-8568, Japan; ikezaki@ubi.s.u-tokyo.ac.jp (K.I.); masahiro.kuramochi.vw26@vc.ibaraki.ac.jp (M.K.); 5Japan Synchrotron Radiation Research Institute, SPring-8, 1-1-1 Kouto, Sayo 679-5198, Japan; sekiguchi@spring8.or.jp; 6Department of Physics, College of Humanities and Sciences, Nihon University, Sakurajosui 3-25-40, Tokyo 156-8550, Japan; judai@chs.nihon-u.ac.jp; 7Department of Biosciences, College of Humanities and Sciences, Nihon University, Tokyo 156-8550, Japan; saitou.minoru79@nihon-u.ac.jp; 8AIST-UTokyo Advanced Operando-Measurement Technology Open Innovation Laboratory (OPERANDO-OIL), National Institute of Advanced Industrial Science and Technology (AIST), 6-2-3 Kashiwanoha, Chiba 277-0882, Japan

**Keywords:** tubulin dimer, molecular motility, diffracted X-ray tracking, molecular dynamics simulation

## Abstract

Tubulin has been recently reported to form a large family consisting of various gene isoforms; however, the differences in the molecular features of tubulin dimers composed of a combination of these isoforms remain unknown. Therefore, we attempted to elucidate the physical differences in the molecular motility of these tubulin dimers using the method of measurable pico-meter-scale molecular motility, diffracted X-ray tracking (DXT) analysis, regarding characteristic tubulin dimers, including neuronal TUBB3 and ubiquitous TUBB5. We first conducted a DXT analysis of neuronal (TUBB3-TUBA1A) and ubiquitous (TUBB5-TUBA1B) tubulin dimers and found that the molecular motility around the vertical axis of the neuronal tubulin dimer was lower than that of the ubiquitous tubulin dimer. The results of molecular dynamics (MD) simulation suggest that the difference in motility between the neuronal and ubiquitous tubulin dimers was probably caused by a change in the major contact of Gln245 in the T7 loop of TUBB from Glu11 in TUBA to Val353 in TUBB. The present study is the first report of a novel phenomenon in which the pico-meter-scale molecular motility between neuronal and ubiquitous tubulin dimers is different.

## 1. Introduction

Eukaryotic microtubules, which are required to maintain morphology and homeostasis in cells, are also involved in the vesicle membrane transport function essential for the long axonal transport of transmitters in neurons. These adjustments are supported by a number of microtubule-associated proteins (MAPs), including molecular motors, which are capable of regulating the dynamic instability of microtubules [1]. The interaction manner of these MAPs with the microtubules has been generally known to occur primarily through electrostatic interactions of basic amino acids in the tubulin-binding domain (TBD) exposed on the molecular surface of MAPs with the exposed acidic C-terminus (E-hook) of a single tubulin subunit [2,3,4,5]. These studies have focused on its structure and function, based mainly on the electrostatic relationship between tubulin and various MAPs, to investigate the physiological function of microtubules. However, to date, no study has examined the physical features of the tubulin dimer itself, consisting of a combination of tubulin isoforms which have been recently reported to form a large family consisting of many gene isoforms [6].

Microtubules are formed by multiple tubulin isoforms that alter microtubule dynamics, mechanical properties, and intrinsic microtubule properties, such as the recruitment and activity of motor proteins and MAPs [7]. Thus, the significance of the diverse tubulin isoforms lies in regulating the diversity of microtubule properties. Tubulin isoforms are also known to be expressed in a tissue-specific manner and are essential for specialized microtubule functions in sperm, platelets, and neurons [8,9]. One reason why microtubules composed of diverse tubulin isoforms vary in their properties is thought to be the effects of diverse post-translational modifications that differ for each tubulin isoform [7]. However, there have been no studies on the differences in motility between tubulin dimer isoforms, although there are several known examples in other proteins where a slight mutation in an amino acid results in an extreme difference in dynamics [10].

The molecular dynamics (MD) simulation method is used to examine in detail the dynamics involved in protein function on the order of nanoseconds to microseconds. Many MD simulations have been conducted on tubulin molecules. Many of these studies have focused on the binding of drugs to tubulin dimers using MD simulations [11,12,13]. Other studies have investigated the stability of the pre-filament state of tubulin octamers [14] and the conformational stability of tubulin dimers, including β-tubulin bound to GTP or GDP [15]. Many other studies have used molecular dynamics simulations of tubulin dimers, but none have compared differences in dynamics between tubulin dimer isoforms.

Therefore, in the present study, we attempted to verify whether there are differences in the molecular motility of tubulin dimers consisting of tissue-specific isoforms. For this purpose, the molecular motility of characteristic tubulin dimers, including neuronal TUBB3 and ubiquitous TUBB5, was measured using measurable pico-meter-scale molecular motility and diffracted X-ray tracking (DXT) analysis [16,17,18,19], and we found that the motility of neuronal tubulin dimer was lower than that of the ubiquitous one. Additionally, we performed MD simulations of these tubulin dimers, and we suggest the possibility of a difference between the motility of neuronal and ubiquitous tubulin dimers, which may be caused by a change in the significant contact of Gln245 in the T7 loop of TUBB from Glu11 in TUBA to Val353 in TUBB.

## 2. Results

### 2.1. Molecular Motility around the Vertical axis of Neuronal Tubulin Dimers Is Lower Than That of Ubiquitous Tubulin Dimers

Differences in the molecular features of tubulin dimers, consisting of a combination of various tubulin isoforms, are not yet known. Therefore, we focused on the characteristic neuronal and ubiquitous tubulin dimers, including TUBB3 and TUBB5, respectively, and tried to elucidate the differences in the molecular motility of these tubulin dimers using DXT analysis, which is a method that can measure pico-meter-scale molecular motility [16] (Figure 1a). We first performed DXT analysis using tubulin dimers reconstructed from a recombinant protein consisting of a combination of TUBB3-TUBA1A and TUBB5-TUBA1B as subunits of neuronal and ubiquitous tubulin dimers, respectively. Figure 1b shows the SDS-PAGE gel separation image of each tubulin subunit.

Tubulin dimers were synthesized and purified using the *E. coli* expression system, Co-NTA, and a gel filtration column. The molecular motility of neuronal (TUBB3-TUBA1A) and ubiquitous (TUBB5-TUBA1B) tubulin dimers reconstructed from each purified subunit protein, detected as one band (Figure 1b, marked with an arrowhead), was measured using DXT. DXT analysis was performed under the condition that the tubulin dimer was bound perpendicularly to the substrate of the thin gold film with absorbed cobalt ions in the instrument’s observation cell via the N-terminal His tag of TUBA (Figure 1e). In the DXT experiment, based on the diffraction spot data obtained from the measurements, the plot of the time interval Δt against the mean squared displacements (MSD) of the two components, tilting (θ) and twisting (χ), was used to discuss the molecular mobility and the diffusion process of such mobility [20]. Here, we present a plot of the MSD of the tilting (θ) and twisting (χ) motions (Figure 1c,d). Figure 1c,d show a decrease in the MSD value in the direction of rotation of the chi and theta axes in the neuronal tubulin dimer (TUBB3-TUBA1A). This result indicated that the molecular motility in the vertical axis rotation direction of the neuronal tubulin dimer (TUBB3-TUBA1A) was lower than that of the ubiquitous tubulin dimer (TUBB5-TUBA1B).

Interestingly, similar results were also observed in purified endogenous neuronal and ubiquitous tubulin dimers without MAPs, including motor proteins affinity-purified with antibodies against TUBB3 and TUBB5 from brain and liver tissues, respectively. DXT analysis using endogenous tubulin dimers revealed that the MSD value in the theta direction was almost unchanged in the time order of 400 ms as molecular motility in the neuronal tubulin (TUBB3) dimer (Appendix A), whereas the MSD value in the chi direction decreased in the time order of 400 ms as it was in the neuronal tubulin dimer (Appendix A).

These results suggest that the molecular motility of neuronal tubulin dimers is lower than that of the ubiquitous tubulin dimers.

### 2.2. In MD Simulations, the Motility of Neurons and Ubiquitous Tubulin Dimers Was Consistent with the Results from DXT Analysis

In this study, 1 μs MD simulations were conducted five times on the tubulin dimers TUBB3-TUBA1A and TUBB5-TUBA1B. Both dimers comprise a core region and a C-terminal tail. The core regions of α-tubulin and β-tubulin were defined as Met1-Gly436 and Met1-Ala427, while the C-terminal tail regions were Val437-Tyr451 and Leu428-Tyr451, respectively. The RMSD values for the Cα atoms situated in the core region of each tubulin exhibited a range between 1 and 2 Å, signifying the stability of the core region (Appendix A). This stability was further underscored by the Cα-RMSF values from the simulations (Appendix A). The Cα-RMSF analysis showed flexibility primarily in the loop regions, exceeding 2 Å, while the rest of the structure fluctuated under 2 Å. However, relative RMSDs of TUBA and TUBB in each dimer ranged between 2 and 9 Å, suggesting a positional displacement between them (Figure 2a,b). Furthermore, by comparing the structure at points of significant displacements in the RMSD plots to the initial structure (Figure 2c,d) and juxtaposing the final structure with the initial one (Figure 2e,f), it was inferred that these shifts in relative positions represent counterclockwise deviations from the initial structure.

Subsequently, a principal component analysis (PCA) was conducted to examine the relative motions of the α- and β-tubulin dimers (Figure 3). In the PCA plot of TUBB3-TUBA1A (Figure 3a), a unimodal distribution was observed, characterized by a peak situated near (0,0) and exhibiting a presence frequency of 3.0%. Conversely, TUBB5-TUBA1B (Figure 3b) displayed a distribution with a flattened peak and a presence frequency of less than 1.5%. These observations suggest that TUBB5-TUBA1B is inclined to adopt more diverse states concerning the relative positions of alpha- and beta-tubulin compared to TUBB3-TUBA1A. Projection of the motion vectors of PC1 and PC2 onto the tubulin dimer structure showed associated twisting (Figure 3c) and tilting (Figure 3d) motions. The specific motion directions, represented by blue and orange arrows in Figure 1e, had contribution ratios of 0.451 and 0.265, respectively, signifying that tilting and twisting are the primary relative motions between TUBB and TUBA in the tubulin dimer.

In the distribution of contact frequency between TUBA and TUBB, we observed a shift towards a lower contact frequency in TUBB5-TUBA1B compared to TUBB3-TUBA1A (Figure 4). This shift signifies diminished interactions between the interfaces, leading to enhanced mobility. These observations align coherently with the outcomes derived from the DXT experiments.

These contacts were calculated from all the MD simulations for TUBB3-TUBA1A and TUBB5-TUBA1B. In the present study, the contact is defined as heavy atoms less than 4 angstroms.

### 2.3. Neuronal Tubulin Dimers Have More TUBA-TUBB Interface Contacts Than Ubiquitous Ones

We compared the contact frequencies of both tubulin dimers at the TUBB-TUBA interface (Figure 5).

In the TUBA interface, residues Tyr224 and Gln11 displayed fewer contacts in TUBB5-TUBA1B compared to TUBB3-TUBA1A (Figure 5a,b). Similarly, within the TUBB interface, residue Arg46 exhibited reduced contact frequency in TUBB5-TUBA1B relative to TUBB3-TUBA1A (Figure 5c,d). Among these residues, Tyr224 (TUBA) and Gln11 (TUBA) exhibited a high contact frequency with Gln245, Leu246, and Asn247 of TUBB (Appendix A). Moreover, Arg46 (TUBB) primarily exhibited a high contact frequency with Thr73 (Appendix A).

Gln11 and Tyr224 of TUBA are located near the GTP of TUBA and exhibited a high contact frequency with GTP in MD simulations, suggesting that these residues form very stiff structures (Appendix A). In contrast, the amino acids Gln245, Leu246, and Asn247 of TUBB, which were in contact with Gln11 and Tyr224 of TUBA, were contained in a single loop called the T7 loop (Appendix A). The structure of this loop region was observed to be divided into two primary states (Figure 6c). In one state, the T7 loop is oriented toward TUBA (State A); in the other state, the T7 loop flips and faces TUBB itself (State B). Particularly, in these two states, Gln245 in the T7 loop has different contact partners: TUBA (Tyr224 or Gln11, State A) and TUBB (Val353, State B).

To elucidate the populations of States A and B in neuronal and ubiquitous tubulin dimers, we analyzed the distribution of distances between residues Gln245 (TUBB) and Gln11 (TUBA) as well as between Gln245 (TUBB) and Val353 (TUBB) (Figure 6a,b). For the distance between Gln245 (TUBB) and Gln11 (TUBA), TUBB3-TUBA1A predominantly exhibited around 4Å (Figure 6a). Conversely, TUBB5-TUBA1B demonstrated nearly equal distribution at both 4Å and 12Å, with the 4Å distribution being less frequent than in TUBB3-TUBA1A (Figure 6a).

For the distribution of distances between Val353 (TUBB) and Gln245 (TUBB), while TUBB3-TUBA1A primarily exhibits distances of around 10Å and 12Å, TUBB5-TUBA1B predominantly shows distances of around 4Å (Figure 6b). Based on these results, it is evident that TUBB3-TUBA1A predominantly exhibits State A, while, conversely, TUBB5-TUBA1B primarily features State B. This observation is further substantiated by analyzing the time course of distances depicted in Figure 6a,b for each MD simulation run and by the calculated presence ratios of State A and State B determined in each individual run (Appendix A).

## 3. Discussion

In the present study, we found that tubulin dimers composed of tubulin isoforms from diverse tissues—neuronal and ubiquitous—manifest distinct variations in motility. This phenomenon was supported by the DXT analysis of reconstructed tubulin dimers of TUBB3-TUBA1A and TUBB5-TUBA1A recombinant proteins (Figure 1) and endogenous tubulin dimers, including neuronal TUBB3 and ubiquitous TUBB5 (Appendix A).

To validate the differences in motility between neuronal and ubiquitous tubulin dimers as revealed by DXT analysis, we conducted MD simulations. The results from Cα-RMSD, Cα-RMSF, and PCA analyses demonstrated that MD simulations reproduced the findings of DXT. We discerned that the key factor driving these differences is the varied contact frequency at the interface of TUBA and TUBB. The main factor contributing to this observed discrepancy lies in the differing contact frequencies at the interface between TUBA and TUBB. Our study specifically highlights differences in the contact frequencies of interface residues in both neuronal and ubiquitous tubulin dimers, notably Tyr224 and Gln11 in TUBA and Arg46 in TUBB. These residues possess either polar or charged side chains.

MD simulations have demonstrated that the differences in contact at the tubulin interfaces are primarily attributed to the movement of the T7 loop of TUBB within the tubulin dimer interface. The T7 loop of TUBB is known to contribute to interactions with drugs acting at the tubulin dimer interface, such as colchicine [22]. Moreover, it exerts influence over the morphology of tubulin dimers, affecting both their curvature and straightness [23]. Furthermore, recent MD simulations of tubulin dimers have revealed the flip motion of the T7 loop of TUBB [24]. From the findings above, it is reasonable to infer that the T7 loop, being crucial for the conformational changes in tubulin dimers, plays a significant role in the observed differences in tubulin dimer kinetics in this study. So, what could be the reason for the differences in the behavior of the T7 loop between neuronal and ubiquitous tubulin dimers? We believe that the three mutation sites between TUBB3 and TUBB5 located near the T7 loop might explain the observed differences in its mobility.

Three mutations were identified near the region of the T7 loop responsible for the differences in contact frequency between TUBB3-TUBA1A and TUBB5-TUBA1B at the TUBA-TUBB interface (Appendix A). The specific residues include 45 (TUBB3: Glu, TUBB5: Asp), 239 (TUBB3: Ser, TUBB5: Cys), and 351 (TUBB3: Val, TUBB5: Thr). The impact of these residue variations on contact frequency is elaborated upon below.

For residue 45, TUBB3-TUBA1A has glutamic acid while TUBB5-TUBA1B has aspartic acid. These residues are positioned next to the N-terminal side of Arg46, where a difference in contact frequency between the two tubulin dimers was observed. Though both residues carry a negative charge, they vary by one carbon in their side chains. This suggests distinct interactions with the positively charged Arg side chain, potentially influencing the conformational stability of Arg46.

Residues 239 and 351, situated near the T7 loop region, influence the contact frequency difference between TUBA and TUBB. Notably, position 351 is adjacent to Val353, observed to interact with the T7 loop (Figure 6a,b). While TUBB3-TUBA1A has a hydrophobic valine at this position, TUBB5-TUBA1B has a hydrophilic threonine. This likely causes a variation in interactions with the loop’s main chain carbonyl and amide groups, potentially contributing to stabilization in the TUBB side for TUBB5-TUBA1B.

Additionally, as depicted in Appendix A, amino acid mutations at the interfaces of the aforementioned neuronal and ubiquitous tubulin dimers do not induce alterations in the distribution of amino acid types (positive, negative, polar, and hydrophobic). Consequently, it is improbable that the displacement of these amino acids would directly instigate a modification in the interface contacts of both dimers.

Next, we tried to compare the results of the DXT experiment with the results of the MD simulations obtained in this study, focusing on the diffusion constants of the two modes of rotational motion observed in the DXT experiment. Using the method shown in Appendix A, we measured tilting and twisting motions from the MD data, made MSD plots (Appendix A), and determined diffusion constants (Appendix A). We then compared these results with the DXT experiment data.

In the diffusion constants derived from MD simulations, it was demonstrated that the constant for twisting exceeded that for tilting in both neuronal and ubiquitous tubulin dimers. This aligns with the trend observed in the diffusion constants acquired from DXT experiments. Furthermore, a comparison of the diffusion constants for the twisting and tilting motions of the neuronal and ubiquitous tubulin dimers was made. The findings indicate that the diffusion constants for twisting are greater in ubiquitous tubulin dimers compared to the neuronal, aligning with the observations from the DXT experiments. However, a deviation was noted where the diffusion constants for twisting motions were marginally higher in the neuronal than in the ubiquitous, a discrepancy from the results of the DXT experiment. This could be due to the following reasons.

In the DXT experiments, the tubulin dimer displayed two unique bound states when it bound to the substrate. In the first state, the axis that passes through TUBB and TUBA was aligned parallel to the vertical axis (Mode one, see Appendix A). On the other hand, in the second state, this axis was positioned perpendicular to the vertical axis (Mode two, refer to Appendix A). In the DXT experiments, the majority of tubulin dimers predominantly adopted mode one, with only a minority exhibiting mode two. Intriguingly, the twisting motion identified in Mode 1 was interpreted as a tilting motion in Mode 2. Consequently, a discrepancy between the experimental and calculated values of tilting motions is likely to emerge. This phenomenon is particularly prominent in the case of the ubiquitous tubulin dimer, characterized by substantial twisting motions.

In addition, the diffusion constants obtained from the MD simulations and DXT experiments differed by substantial orders of magnitude. In the DXT experiments, the tubulin dimers were bound to the substrate, which significantly constrained their motion compared with their free state in our MD simulations. In addition, in the DXT experiments, the tubulin dimer was bound to a very large gold nanocrystal (40–80 nm) compared to the tubulin dimer, with a long axis of approximately 10 nm. Thermal fluctuations due to Brownian motion, which are different in scale from the motion in the protein molecule, are also considered to affect the observation [25] and are the reason for the difference in the order of the diffusion constants between DXT and MD. Kawashima et al. investigated the mobility of peptide molecules under constrained conditions in DXT using replica exchange molecular dynamics simulations. They found that the stability of the molecular conformation was the same as that of the free molecules, but the mobility was slightly constrained [25]. Although it is a future task to make the difference between the diffusion constants obtained from DXT and MD as small as possible, it is clear that the intramolecular motion of tubulin dimer from the neuronal (TUBB3-TUBA1A) is suppressed in the torsional (χ) direction compared to the ubiquitous (TUBB5-TUBA1B).

In the context of DXT experiments involving endogenous tubulin dimers, which include neuronal TUBB3 and ubiquitous TUBB5 (Appendix A), the His tag was absent. Computational studies on the binding affinity of gold and amino acids have been performed. The stability of gold interactions with amino acids other than cysteine, which form covalent bonds, has been studied and shown to be stabilized by interactions with the main and side chains and by water-mediated interactions [26]. In addition, a systematic study of the binding affinity of amino acids to gold atoms has been conducted based on quantum chemical calculations by Buglak and colleagues, which showed that the following series of amino acids are particularly likely to bind to gold atoms [27]: Cys(−H+) > Asp(−H+) > Tyr(−H+) > Glu(−H+) > Arg > Gln, His, Met.

These amino acids were present throughout both the TUBB3-TUBA1A and TUBB5-TUBA1B tubulin dimers, and the distribution of amino acids around the surface that could interact with gold was also present throughout the dimers (Appendix A). Considering the above, various orientation modes can be assumed in the system used in the DXT experiment of endogenous tubulin dimers. Although the orientation mode may be different between endogenous and recombinant tubulin dimers, both of the diffusion constants obtained from the DXT experiments were of approximately the same order of time (Appendix A). It suggests that molecular motions similar to those observed in recombinant tubulin dimers are also observed in endogenous ones. Furthermore, the diffusion constant, derived from the MSD plots (Appendix A) by combining tilting (θ) and twisting (χ) (Appendix A), which is a metric employed for comparison in DXT experiments in systems with indeterminate orientations, revealed that the motility of the neuronal tubulin dimer was marginally more restrained than that of the ubiquitous one. This observation is analogous to the variance noted in the DXT experimental results for the recombinant tubulin dimers. These results suggest that the tubulin dimer motions observed in the DXT experiments in this study were independent of differences in the mode of orientation of the substrate and other factors. In addition, differences in motility similar to the results of recombinant tubulin dimers observed in the Native system suggest that differences in motility between isoforms are characteristic of actual in vivo tissues. Therefore, assuming that this is true, the tubulin dimers of neuronal and ubiquitous tissues with the TUBB3-TUBA1A and TUBB5-TUBA1B isoforms may have specific kinetics, and the neuronal and ubiquitous tubulin dimers may have unique kinetics. In the future, we plan to conduct more detailed experiments to compare the motility of native and mutant tubulin dimers.

In this study, we performed MD simulations for each of the two tubulin dimers for a total of 5 μs. Our MD simulations were shorter than the DXT experiments. However, running MD simulations on the DXT timescale would not change the effects of the residues involved in the contact difference, as revealed in our simulations, unless the dimer interface state is completely altered.

Taken together, our results reveal that the molecular motility of neuronal tubulin dimers is lower than that of ubiquitous tubulin dimers. Furthermore, we executed a series of molecular dynamics (MD) simulations, totaling 10 microseconds, for both neuronal and ubiquitous tubulin dimers. These simulations reveal that the differences in interactions between α- and β-tubulin result from the varied motility of the T7 loops of β-tubulin in neuronal and ubiquitous tubulin dimers. We believe these findings are key to understanding the differences in motility among these tubulin isoforms. This result is the first to demonstrate a difference in physical motility between tubulin dimers. By altering the stoichiometric ratio of tubulin isoforms, it has been demonstrated that changes occur in the dynamics of the resulting microtubules [28]. Additionally, it is established that a specific isoform is overexpressed in cancer cells, resulting in heightened dynamics of microtubules. This is suggested as one of the reasons for the resistance of cancer cells to anti-microtubule drugs [29]. Despite the established understanding of alterations in microtubule behavior influenced by isoforms, the underlying principles governing how tubulin isoforms, translational modifications, and additional factors impact the physical attributes of microtubules continue to remain elusive [7]. The valuable results obtained in this study may shed light on the phenomena associated with differences in tubulin isoforms. To clarify why the physical motility of tubulin dimers in neurons is low and how this novel phenomenon contributes to various physiological functions of microtubules such as vesicular transport, further study on the vesicular transport system using the dimer and its constituent microtubules including MAPs is needed.

## 4. Materials and Methods

### 4.1. Materials

All chemicals and reagents were purchased from Sigma-Aldrich (St. Louis, MO, USA), FUJIFILM Wako Pure Chemical (Osaka, Japan), and Invitrogen (Carlsbad, CA, USA), except for *E. coli* culture reagents, which were purchased from Nacalai Tesque (Kyoto, Japan), unless otherwise noted.

### 4.2. Preparation of Tubulin Dimer

For the synthesis and purification of the recombinant tubulin dimer, Escherichia coli expression constructs for tubulin isoform genes (TUBB3, TUBB5, TUBA1A, and TUBA1B) were constructed using the pDEST17 vector of the gateway expression system to produce His-tagged proteins. Tubulin isoforms were synthesized in *E. coli* strain BL21 (DE3) pLysS. Cells were harvested using centrifugation, suspended in lysis buffer (8M Urea, 20 mM Tris-HCl (pH 8.0), 300 mM NaCl), and disrupted by sonication. The lysate was centrifuged at 20,400× *g* for 60 min to pellet cell debris. The supernatant was then subjected to a TALON Co2+ affinity resin. The bound proteins were eluted using an elution buffer (8M Urea, 20 mM Tris-HCl (pH 8.0), 300 mM NaCl, and 500 mM imidazole). Tubulin isoform proteins were purified with size exclusion chromatography at 4 °C using a HiLoad 16/600 Superdex75 pg with buffer (100 mM MES pH 6.4, 150 mM KCl, and 1 mM CaCl2). The fractions containing each protein isoform were collected. After purified recombinant tubulins were validated with coomassie brilliant blue staining with 13% SDS-PAGE separation, the neuronal and ubiquitous tubulin dimers were reconstituted at 4 °C in buffer (100 mM MES-pH 6.4, 150 mM KCl, 1 mM CaCl2, PI cocktail), containing a combination of TUBB3-TUBA1A and TUBB5-TUBA1B, respectively. To prepare the endogenous tubulin dimer, pure tubulin without TIPs, isolated from tissues according to a previous report [30,31], was subjected to protein-G column conjugation with antibodies of TUBB3 or TUBB5 using bis suberate disodium salt to obtain tubulin dimers containing TUBB3 or TUBB5.

### 4.3. Sample Preparation for Diffracted X-ray Tracking

Sample substrates for DXT were prepared by depositing a well-adherent vapor-deposited gold thin film (8 nm thick) on Kapton film (25 microns thick), which has excellent X-ray durability. Cobalt ions (1 mM) were adsorbed onto the gold substrate surface to adsorb His-tagged tubulin. In this study, tubulin had a reaction time of 1 h with the DXT substrate. The gold film on the substrate was amorphous; therefore, the diffraction spots did not overlap with the gold nanocrystals to be labeled. Next, tubulin adsorbed on the substrate reacted with the gold nanocrystals. The reaction time was approximately 5 min. High-quality gold nanocrystals were prepared by epitaxial growth on KCl (111) single crystals.

### 4.4. Diffracted X-ray Tracking (DXT)

In this study, DXT was performed with SPring-8 BL40XU using X-rays with an energy width of 14.0–16.5 keV and a photon flux of 1013 photons/s. The beam size of the incident X-rays was adjusted to 50 µm in diameter with a pinhole slit. Diffraction images of the gold nanocrystals were recorded using an X-ray image intensifier (diameter 100 mm, v7339P, Hamamatsu Photonics, Hamamatsu, Japan) and a CMOS camera (Phantom V2511, Vision Research, Wayne, USA). The distance between the detector and the sample was 50 mm. The time division for this measurement was 50 ms. The measurements were performed on the same sample at 6 × 6 locations with DXT measurements between 15 s of X-ray irradiation at each measurement position. No X-ray damage was observed during this irradiation time, and the total number of diffraction spots obtained in the DXT measurements was approximately 100–200 spots per sample.

### 4.5. Modeling of Tubulin Dimers

A tubulin dimer is formed by two distinct proteins: α-tubulin (TUBA) and β-tubulin (TUBB). These proteins exhibit similar secondary structural elements, primarily consisting of ten beta strands, twelve alpha helices, and seven loops [32], as depicted in Appendix A. Notably, both α-tubulin and β-tubulin can bind guanine nucleotides, a critical factor in microtubule elongation. While α-tubulin has an affinity for GTP, β-tubulin predominantly binds GDP in its isolated dimeric state (refer to Appendix A for details).

In this study, we examined neuronal (TUBB3-TUBA1A) and ubiquitous (TUBB5-TUBA1B) tubulin dimers. At the time of our modeling (October 2017), the tubulin isoforms for which crystal structures had been resolved included TUBA1A, TUBA1B, TUBA4B/8, TUBB2A, and TUBB2B. The structures of the isoforms TUBB3 and TUBB5, used in this study, were not available. In this study, we utilized PDB_ID: 3RYC (TUBA1B-TUBB2A, resolution 2.1Å) as the primary template for homology modeling. This crystal structure was the highest resolution structure available at the time among those containing TUBA1A or TUBA1B in tubulin dimers. Subsequently, we conducted modeling of TUBA1A-TUBB3 and TUBA1B-TUBB5 using the homology modeling program MODELLER [33]. The sequence identities between the TUBA in 3RYC and the TUBA1A and TUBA1B used in this study were 99.56% and 100%, respectively. Additionally, the sequence identities between the TUBB in 3RYC and the TUBB3 and TUBB5 used in this study were 92.13% and 94.82%, respectively. The sequence identity between TUBA1A and TUBA1B used in this study was 99.56%, and between TUBB3 and TUBB5, it was 92.57%.

The details of the template structures used for homology modeling are as follows: the A chain and B chain of PDB_ID: 3RYC were used as the primary templates for TUBA and TUBB, respectively. Parts of the structures of PDB_ID: 2E4H and PDB_ID: 4I4T were used as templates for the missing regions of TUBA and the C-terminal tail region of TUBA, respectively. In Appendix A, we have depicted the sequence alignment of the templates used in the homology modeling. GTP and GDP were bound to TUBA and TUBB of the modeled dimers, respectively. In addition, Mg ions were placed between the beta and gamma phosphates in the GTP bound to TUBA. The crystal waters in the 3RYC structure were contained in the modeled TUBA-TUBB dimer structures.

The structures of TUBB3-TUBA1A and TUBB5-TUBA1B, obtained through homology modeling, underwent quality assessment using MolProbity [34]. Each structure received a MolProbity Score of 1.86 and 1.94, respectively, which are lower than the score of 2.02 for the template structure 3RYC. A lower MolProbity score indicates higher structural quality. Consequently, both structures generated through modeling exhibited superior quality compared to the crystal structure employed as the primary template. The Cα-RMSD value between the modeling structures is 0.344Å. In Appendix A, we have depicted the modeling structures and the template structure from PDB_ID 3RYC. Furthermore, in both modeling structures, the T7-loop of TUBB was in State A, as depicted in Figure 6. From these results, it is evident that the homology modeling performed in this study successfully yielded high-quality and highly similar structures of TUBB3-TUBA1A and TUBB5-TUBA1B.

### 4.6. Molecular Dynamics Simulation

MD simulations were performed using GROMACS ver2016.1 [35,36,37,38] with an amber99SBnmr1-ildn force field [39]. The modeled structures of the TUBB3-TUBA1A and TUBB5-TUBA1B dimers were used as the initial structures. These dimers were embedded in TIP3P water [40], and sodium ions were added to neutralize the system. The simulation system was an NPT ensemble, and the temperature and pressure were set to 300 K and 1atm controlled by the V-rescale [41] and Parrinello-Rahman methods [42], respectively. The Partivle mesh Ewald (PME) [43] was used to calculate the Coulomb interaction. The calculation of the van der Waals (vdw) interactions utilized a short-range van der Waals cutoff of 1.0 nm. All bonds involving hydrogen atoms were constrained using the Linear Constraint Solver (LINCS) algorithm [44]. In addition, the step length in the MD simulation was set to 0.002. ps. The protonation state of the dimers was determined using ProPKa, assuming a state at pH 7.0 [45,46]. In this study, 5 runs of 1 μs MD simulations were performed independently for each dimer, and snapshots were obtained every 100 ps during the MD simulations. These simulations were performed using CRAY XC50 and TSUBAME3.0.

## Figures and Tables

**Figure 1 ijms-24-15423-f001:**
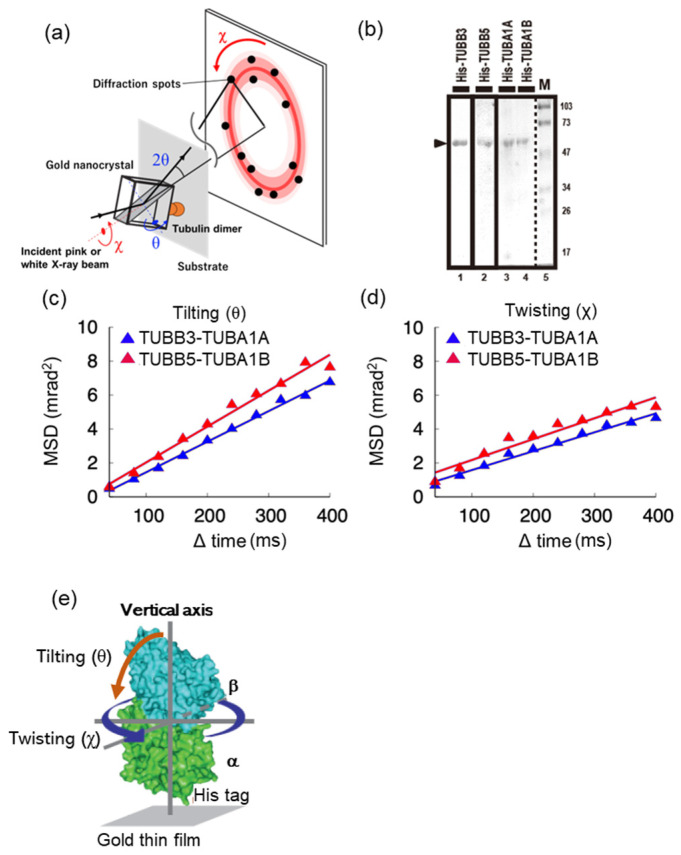
Diffracted X-ray tracking (DXT) analysis using tubulin dimers reconstructed from purified recombinant subunit proteins. (**a**): Schematic diagram of DXT analysis. DXT measurements realize dynamic measurements of single molecules labeled with nanocrystals with time-resolved tracking of the motion of 2D diffraction spots defined by θ and χ. (**b**): Purification of recombinant tubulin subunits. Tubulin subunits of neuronal (TUBB3-TUBA1A) and ubiquitous (TUBB5-TUBA1B) tubulin dimers were synthesized by using the *E.coli* expression system and purified with Co-NTA resin. Purified recombinant tubulins were validated with 13% SDS-PAGE, and each tubulin dimer was reconstructed in buffer. Purified tubulin proteins are indicated with an arrowhead. (**c**): Tilting motions of DXT measurement of reconstructed tubulin dimers. (**d**): Twisting motions of DXT measurement of reconstructed tubulin dimers. In panels (**c**,**d**), there are lines drawn using χ-square linear fitting. (**e**) Schematic of DXT analysis using the reconstructed tubulin dimer. The two mobility dimensions in the DXT, twisting (χ) and tilting (θ), indicate the mobility around the vertical axis (blue arrow) and tilting relative to the vertical axis (orange arrow), respectively.

**Figure 2 ijms-24-15423-f002:**
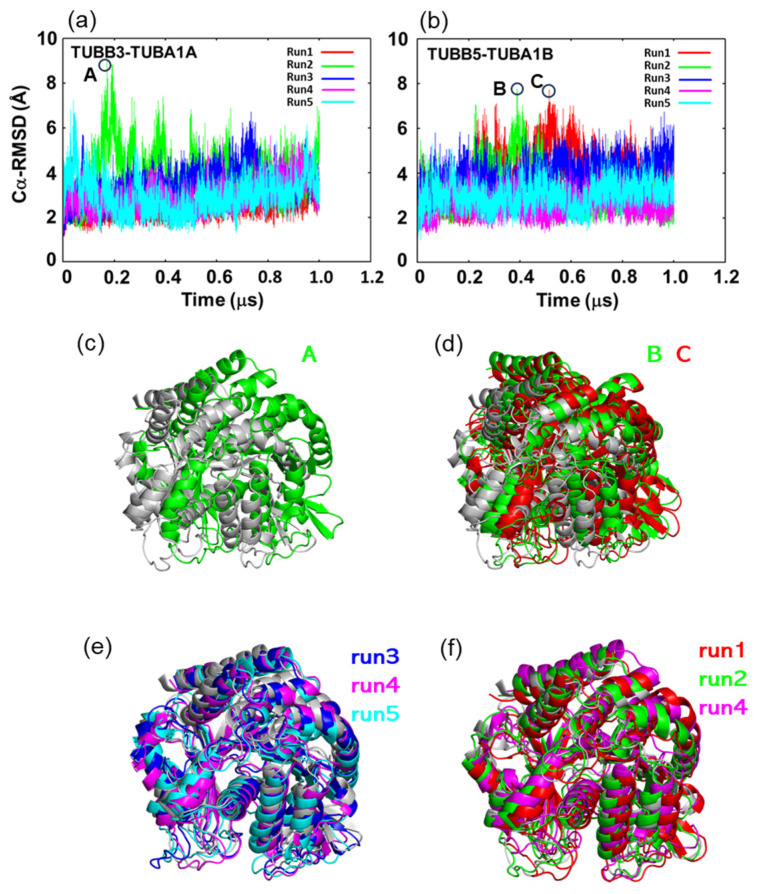
Relative motion of tubulin molecules in TUBB-TUBA complexes. (**a**) Relative Cα-RMSD of TUBB3-TUBA1A. (**b**) Relative Cα-RMSD of TUBB5-TUBA1B. In these panels, frames with especially large RMSD values are encircled and labeled as A, B, and C. These relative RMSD values represent the Cα-RMSD of TUBB molecules when the TUBB-TUBA complexes were fitted with the core region of TUBA molecules. The Cα-RMSD were calculated using Cα atoms of the tubulin core region, defined as 1-436 (TUBA1A, TUBA1B) and 1-427 (TUBB3, TUBB5), respectively. (**c**) Comparison between the initial structure and the snapshot with a large RMSD value labeled as A in panel (**a**). (**d**) Comparison between the initial structure and the snapshot with large RMSD values labeled as B and C in panel (**b**). (**e**) Comparison between the initial and final structures of runs 3, 4, and 5 of the MD simulations of TUBB3-TUBA1A. (**f**) Comparison between the initial and final structures of runs 1, 2, and 4 of the MD simulations of TUBB5-TUBA1B. Panels (**c**–**f**) display TUBB structures with TUBA superimposed on the initial structure. In panels (**e**,**f**), three final structures with the lowest RMSD values in MD simulations of TUBB3-TUBA1A and TUBB5-TUBA1B were selected. In panels (**c**–**f**), molecular graphics were drawn using molecular graphics program PyMOL [21].

**Figure 3 ijms-24-15423-f003:**
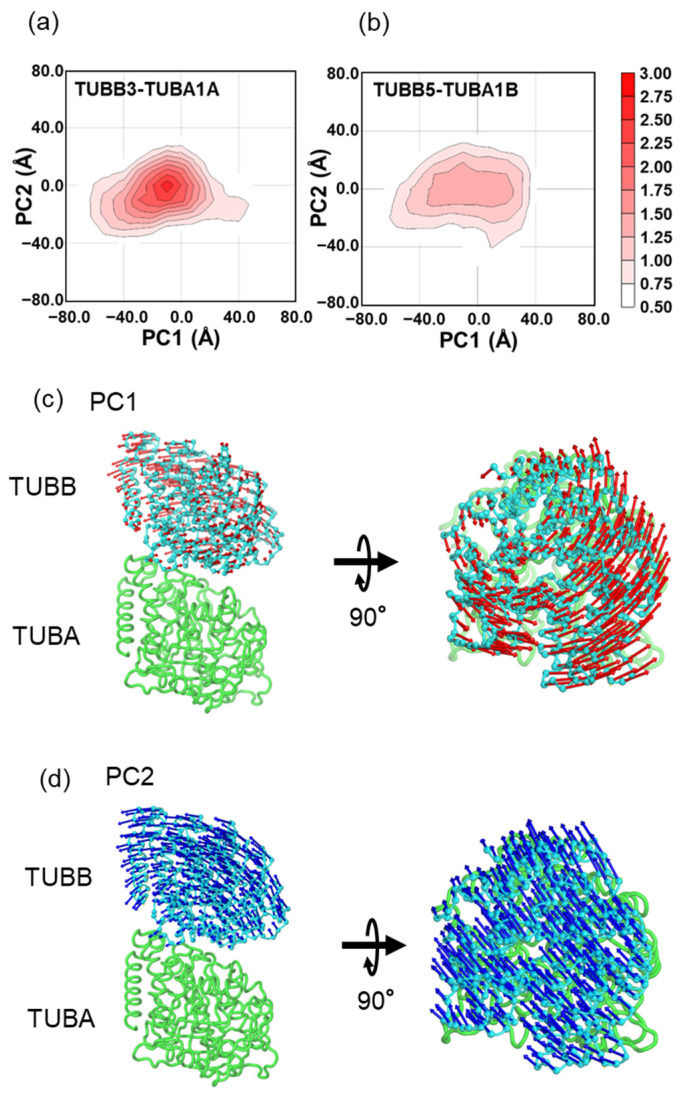
The results of principal component analysis (PCA). (**a**) The PCA map of TUBB3-TUBA1A. (**b**) The PCA map of TUBB5-TUBA1B. The principal axes in these figures are those obtained from the trajectories of all MD simulations in TUBB3-TUBA1A and TUBB5-TUBA1B. (**c**) Vector components of PC1 projected onto the TUBA-TUBB structure. (**d**) Vector components of PC2 projected onto the TUBA-TUBB structure. In these figures, the relative motion of Cα in the core region of TUBB, when fitted with the Cα of the core region of TUBA, is projected onto the PC1-PC2 plane, and the structural occupancy on the PC1-PC2 plane is shown as a white-to-red gradation. See the legend in Figure 2 for the definitions of the core regions of TUBA and TUBB. In panels (**c**,**d**), the vector components are indicated by red and blue arrows, respectively, and molecular graphics were drawn using molecular graphics program PyMOL [21].

**Figure 4 ijms-24-15423-f004:**
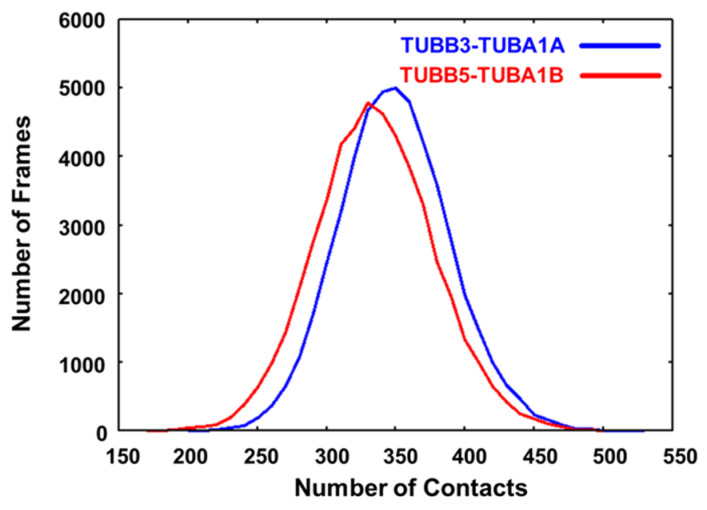
The distribution of the number of contacts at the TUBB-TUBA interface.

**Figure 5 ijms-24-15423-f005:**
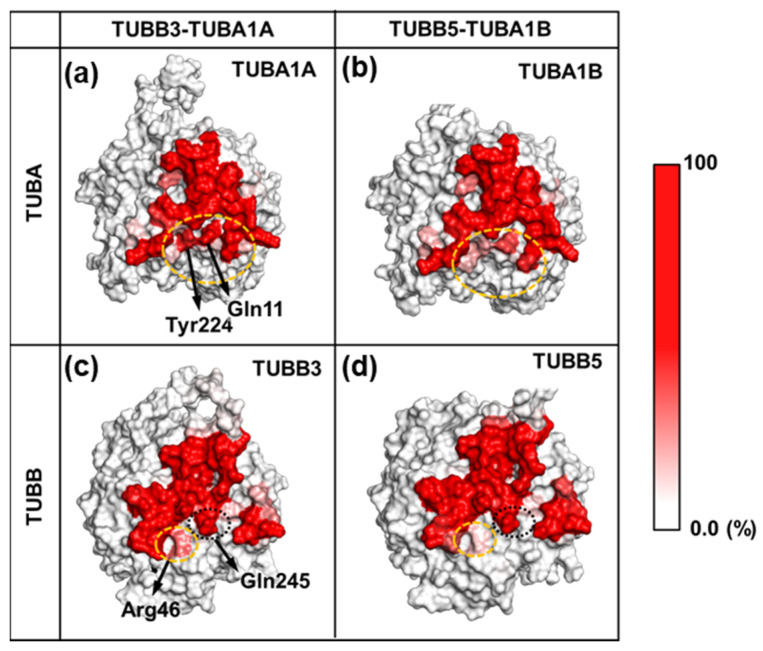
Comparison of contact frequencies at the TUBB-TUBA interface. (**a**) Contact frequencies at the TUBB-TUBA interface for TUBA1A. (**b**) Contact frequencies at the TUBB-TUBA interface for TUBA1B. In panel (a) and (b), the regions of residues Tyr224 and Gln11 with large differences in contact frequency between TUBA1A and TUBA1B are circled in orange dotted line. (**c**) Contact frequencies at the TUBB-TUBA interface for TUBB3. (**d**) Contact frequencies at the TUBB-TUBA interface for TUBB5. In panel (c) and (d), the residue Arg46, which shows a large difference in contact frequency between TUBB3 and TUBB5, is circled by an orange dotted line. In addition, Gln245, which had a large contact frequency with Tyr224 and Gln11 in TUBA, is circled by a black dotted line. These results were obtained from the MD simulations of the TUBB3-TUBA1A and TUBB5-TUBA1B complexes. In the present study, contact is defined as heavy atoms less than 4 angstroms. Here, the contact frequency is the percentage of frames with contacts among all simulations. A white-to-red gradient was used to indicate the contact frequency, and molecular graphics were drawn using molecular graphics program PyMOL [21].

**Figure 6 ijms-24-15423-f006:**
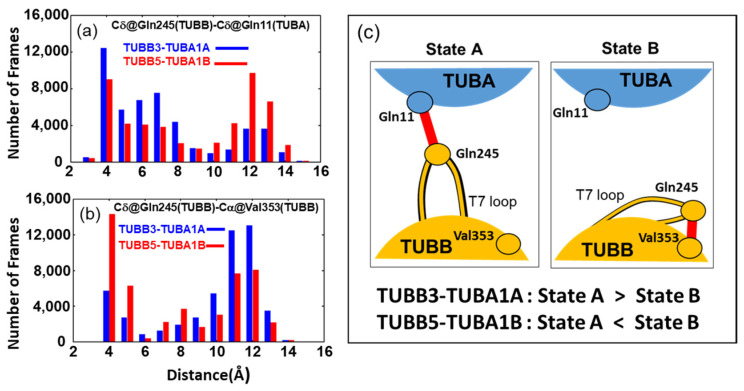
Distribution of the two states of the T7 loop. (**a**) Distribution of the distance between the Cδ of Gln245 in the TUBB T7 loop and the Cδ of residue Gln11 in TUBA. (**b**) Distribution of the distance between the Cδ of Gln245 in the TUBB T7 loop and the Cα of residue Val353 in TUBB. These distributions were obtained from all trajectories of five MD simulations for each of TUBB3-TUBA1A, TUBB5-TUBA1B. (**c**) Schematic of the two states observed in the TUBB-TUBA heterodimer (State A and State B) and the distribution of each state in each heterodimer.

## Data Availability

The data used to support the findings of this study are included in the article and Appendix A.

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
