# Peer review of "Comparison of the Molecular Motility of Tubulin Dimeric Isoforms: Molecular Dynamics Simulations and Diffracted X-ray Tracking Study"

_ijms, 2023, doi:10.3390/ijms242015423_

Round 1
Reviewer 1 Report
In this paper, the authors detected a difference in the molecular motility of neuronal and ubiquitous tubulin dimers using diffracted X-ray Tracking analysis, and MD simulation of these tubulin dimers characterized conformational dynamics of the dimer interface, suggesting a key factor that causes the difference in the motility. While this paper seems to be interesting to the readers of International Journal of Molecular Sciences, there are several concerns in how to show the results, as noted below. I think this paper needs to be revised to correctly address these points.
(Comment 1) Fig1C and D, and Fig. S11:
What is the unit of X-axis? (ms?) (In Fig.S8, the unit is described to be “ns”.)
(Comment 2) Fig.3 C and D:
I think a conformational difference in the twisting and tilting motions is hard to be understandable for readers. (They appear to be similar.)
(Comment 3) Fig.5:
The following sentence was repeated twice. “Here, the contact frequency is the percentage of 200 frames with contacts among all simulations.” Also, I think Gln245 of TUBB should be highlighted in this figure because intermolecular interaction involving this residue is shown to be significantly different between TUBA1A and TUBA1B in Fig.6.
(Comment 4) Regarding functional and biological roles of the low motility of neuronal tubulin dimers or the high motility of ubiquitous tubulin dimers, I found the following statement.
(Line 341) “The low physical motility of neuronal tubulin dimers may affect microtubule function, such as vesicle transport, by increasing the accessibility of motor molecules represented by the KIF protein to the exposed acidic C-terminus of tubulin subunits.”
I think this is too insufficient. In order to develop researches in this field, the authors should more deeply discuss the different motility, incorporating or comparing with information obtained from relevant previous studies.
(Comment) This manuscript seems to be hard to read due to poor English and a complex structure of each sentence. A few examples are described below. I think the authors should check the whole manuscript and try to simplify each sentence/paragraph, removing redundancy.
(Line186) We found that in TUBA, Tyr224 and Gln11 (Figures 5a and 5b), and TUBB, Arg46 (Figures 5c and 5d) had a minor contact frequency in TUBB5-TUBA1B compared to TUBB3-TUBA1A.
->We found that Tyr224 and Gln11 of TUBA less frequently contacted with Arg46 of TUBB in the TUBB5-TUBA1B complex (Figures 5c and 5d) than in the TUBB3-TUBA1A complex (Figures 5a and 5b)?
(Line 217) The distribution at 4 Å is less than in TUBB3-TUBA1A
->TUBB5-TUBA1B exhibited a smaller distribution at 4 Å than TUBB3-TUBA1A?
(Line 267) The diffusion constants tended to be more significant for the twisting motion than for the tilting motion.
->I think this sentence is ambiguous. The difference in the diffusion constants for the twisting motion is more significant than that for the tilting motion?
(Line 271) Comparing the MSD plots of the DXT and MD simulations, the value of TUBB5-TUBA1B was larger than TUBB3-TUBA1A in both plots.
->The MD simulation of the TUBB5-TUBA1B complex provided higher diffusion constants for both the twisting and tilting motions than those of the TUBB3-TUBA1A complex?
Author Response
Reviewer 1
Thank you for your thorough peer review comments on the manuscript. We recognized that our attention to the text and figures was insufficient, so we have focused on refining them in our revisions.
(Comment 1) Fig1C and D, and Fig. S11:
What is the unit of X-axis? (ms?) (In Fig.S8, the unit is described to be “ns”.)
(Answer)
The results of the DXT experiments in Fig. 1C, Fig. 1D, and Fig. S11 (now Fig. S16) are in the unit of ms, while the MSD plots obtained from the MD simulations are described in the unit of ns. Therefore, we have added 'ms' to the x-axis of Fig. 1C, Fig. 1D, and Fig. S16.
(Comment 2) Fig.3 C and D:
I think a conformational difference in the twisting and tilting motions is hard to be understandable for readers. (They appear to be similar.)
(Answer)
We have revised the figures Fig. 3C and 3D to feature more distinct and clear vector arrows. In addition, within Fig. 3C and 3D, we have synchronized the coloring and orientation of the figures that depict the entire tubulin dimer with those in Fig. 1E. Moreover, we have appended information in the manuscript to highlight that the specific rotational directions are illustrated in Fig. 1E, in the section discussing how Fig. 3C and 3D represent tilting and twisting, respectively. (L177-L182).
(Comment 3) Fig.5:
The following sentence was repeated twice. “Here, the contact frequency is the percentage of 200 frames with contacts among all simulations.” Also, I think Gln245 of TUBB should be highlighted in this figure because intermolecular interaction involving this residue is shown to be significantly different between TUBA1A and TUBA1B in Fig.6.
(Answer)
The text of Fig. 5's legend has been revised, and the position of Gln245 in TUBB is now highlighted in Fig. 5. In the figure, there was no difference in the contact frequency of Gln245 (TUBB) between the neuronal and ubiquitous tubulin dimers.
However, the distinction between neuronal and ubiquitous arises in the variations concerning which molecules form hydrogen bonds with Gln245 (TUBB). Therefore, while Fig. 5 does not show a difference in contacts with Gln245 (TUBB), a variation in contact frequency with Gln11 in TUBA, as a binding partner, emerges as a significant alteration affecting the motility differences between these tubulin dimers.
(Comment 4) Regarding functional and biological roles of the low motility of neuronal tubulin dimers or the high motility of ubiquitous tubulin dimers, I found the following statement.
(Line 341) “The low physical motility of neuronal tubulin dimers may affect microtubule function, such as vesicle transport, by increasing the accessibility of motor molecules represented by the KIF protein to the exposed acidic C-terminus of tubulin subunits.”
I think this is too insufficient. In order to develop research in this field, the authors should more deeply discuss the different motility, incorporating or comparing with information obtained from relevant previous studies.
(Answer)
We have incorporated discussions on current understandings regarding isoforms, including reports on the variations in microtubule motility due to changes in isoform ratios, and debates regarding phenomena such as drug resistance to microtubule-targeting drugs in cancer cells, presumed to arise from alterations in isoform ratios. These are added, as follows: “By altering the stoichiometric ratio of tubulin isoforms, it has been demonstrated that changes occur in the dynamics of the resulting microtubules [27]. Additionally, it is established that a specific isoform is over-expressed in cancer cells, resulting in heightened dynamics of microtubules. This is suggested as one of the reasons for the resistance of cancer cells to anti-microtubule drugs [28].” (L387-L392)
Additionally, we touched upon the present ambiguity concerning how precisely changes in isoform ratios elicit alterations in microtubule motility, asserting that the findings from our study offer crucial insights into this matter. We also addressed future research challenges on this topic. These are added, as follows: “To clarify why the physical motility of tubulin dimers in neurons is low and how this novel phenomenon contributes to various physiological functions of microtubule such as vesicular transport, therefore, it is needed to further study on vesicular transport system using the dimer and its constituent microtubules including MAPs.” (L396-L400).
Comments on the Quality of English Language
(Comment) This manuscript seems to be hard to read due to poor English and a complex structure of each sentence. A few examples are described below. I think the authors should check the whole manuscript and try to simplify each sentence/paragraph, removing redundancy.
(Answer)
We have checked the whole manuscript to simplify each sentence/paragraph and eliminate redundancy to enhance the clarity and readability of the manuscript in English. Furthermore, we will answer the following questions from you regarding the manuscript and indicate the corresponding actions taken.
(Line186) We found that in TUBA, Tyr224 and Gln11 (Figures 5aand 5b), and TUBB, Arg46 (Figures 5c and 5d) had a minor contact frequency in TUBB5-TUBA1B compared to TUBB3-TUBA1A.
->We found that Tyr224 and Gln11 of TUBA less frequently contacted with Arg46 of TUBB in the TUBB5-TUBA1B complex (Figures 5c and 5d) than in the TUBB3-TUBA1A complex (Figures 5a and 5b)?
(Answer)
No. There is the two main points we intended to highlight, as followings: (1) Based on the contact analysis results, the contact frequencies of Tyr224 and Gln11 at the TUBA interface were reduced in TUBB5-TUBB1 compared to TUBB3-TUBA1A (Figure 5a and 5b). And (2) The contact frequency of Arg46 in TUBB was also lower in TUBB5-TUBB1 than in TUBB3-TUBA1A (Figure 5c and 5d). We have altered the structure of the discussion in the manuscript and modified the text to ensure clarity on these points. These are added, as follows: “In the TUBA interface, residues Tyr224 and Gln11 displayed fewer contacts in TUBB5-TUBA1B compared to TUBB3-TUBA1A (Figure 5a and 5b). Similarly, within the TUBB interface, residue Arg46 exhibited reduced contact frequency in TUBB5-TUBA1B relative to TUBB3-TUBA1A (Figure 5c and 5d).” (L205-L208)
(Line 217) The distribution at 4 Å is less than in TUBB3-TUBA1A
->TUBB5-TUBA1B exhibited a smaller distribution at 4 Å thanTUBB3-TUBA1A?
(Answer)
Yes, to clarify this sentence, we modified as follows; “Conversely, TUBB5-TUBA1B demonstrates nearly equal distribution at both 4Å and 12Å, with the 4Å distribution being less frequent than in TUBB3-TUBA1A (Figure 6a).” (L225-L227)
(Line 267) The diffusion constants tended to be more significant for the twisting motion than for the tilting motion.
->I think this sentence is ambiguous. The difference in the diffusion constants for the twisting motion is more significant than that for the tilting motion?
(Answer)
No. The point being asserted here is that, "As derived from Table S1, both TUBB3-TUBA1A and TUBB5-TUBA1B display larger diffusion constants for twisting motion as compared to tilting motion." Modifications have been implemented to elucidate this point more clearly. These are added as follows: “In the diffusion constants derived from MD simulations, it was demonstrated that the constant for twisting exceeded that for tilting in both neuronal and ubiquitous tubulin dimers.” (L306-L308).
(Line 271) Comparing the MSD plots of the DXT and MD simulations, the value of TUBB5-TUBA1B was larger thanTUBB3-TUBA1A in both plots.
->The MD simulation of the TUBB5-TUBA1B complex provided higher diffusion constants for both the twisting and tilting motions than those of the TUBB3-TUBA1A complex?
(Answer)
No, the diffusion constant for the tilting motion of TUBB5-TUBA1B, obtained from MD simulations, is slightly smaller than that for TUBB3-TUBA1A. In this section, we discussed the reasons for this outcome. The text to which you referred discussed the values of the MSD plot used to calculate these diffusion constants, compares those of TUBB5-TUBA1B and TUBB3-TUBA1A. However, the key assertion in this section pertains entirely to the diffusion constants. Therefore, to prevent the discussion from becoming overly complicated by introducing considerations of MSD plot values, we limited our discourse solely to discussions of the results related to the diffusion constants. These are added, as follows: “Furthermore, a comparison of the diffusion constants for twisting and tilting motions was made between neuronal and ubiquitous tubulin dimers. The findings indicate that the diffusion constants for twisting are greater in ubiquitous tubulin dimers compared to neuronal, aligning with the observations from the DXT experiments. However, a deviation was noted where the diffusion constants for twisting motions were marginally higher in neuronal than in ubiquitous, a discrepancy from the results of the DXT experiment. This could be due to the following reasons.” (L309-L315)
Reviewer 2 Report

Improve several sections. There are many unclear explanations. Some of the sentences formation is not understanding. Rewrite the sections especially Results, Discussion.
Author Response
Reviewer2
Thank you for your thorough peer review comments on the manuscript. We recognized that our attention to the text and figures was insufficient, so we have focused on refining them in our revisions.
- First, the authors should describe the systems they studied neuronal (TUBB3-TUBA1A) and ubiquitous (TUBB5-TUBA1B) tubulin dimers in the methods section clearly. This should include the 3D snapshot of their secondary structures, GDP/GTP and their alignment to each other.
(Answer)
Per the reviewer's suggestion, we have added the following sentence to the Materials and Methods section.
“A tubulin dimer is formed by two distinct proteins: α-tubulin (TUBA) and β-tubulin (TUBB). These proteins exhibit similar secondary structural elements, primarily consisting of ten beta strands, twelve alpha helices, and seven loops [31], as depicted in Figure S2 and S4. Notably, both α-tubulin and β-tubulin can bind guanine nucleotides, a critical factor in microtubule elongation. While α-tubulin has an affinity for GTP, β-tubulin predominantly binds GDP in its isolated dimeric state (refer to Figure S2 and S3 for details).
In this study, we examined neuronal (TUBB3-TUBA1A) and ubiquitous (TUBB5-TUBA1B) tubulin dimers. At the time of our modeling (Oct. 2017), the tubulin isoforms for which crystal structures had been resolved included TUBA1A, TUBA1B, TUBA4B/8, TUBB2A, and TUBB2B. The structures of the isoforms TUBB3 and TUBB5, used in this study, were not available. In this study, we utilized PDB_ID: 3RYC (TUBA1B-TUBB2A, resolution 2.1Å) as the primary template for homology modeling. This crystal structure was the highest resolution structure available at the time among those containing TUBA1A or TUBA1B in tubulin dimers. Subsequently, we conducted modeling of TUBA1A-TUBB3 and TUBA1B-TUBB5 using the homology modeling program MODELLER [32]. The sequence identity between the TUBA in 3RYC and the TUBA1A and TUBA1B used in this study was 99.56% and 100%, respectively. Additionally, the sequence identity between the TUBB in 3RYC and the TUBB3 and TUBB5 used in this study was 92.13% and 94.82%, respectively. The sequence identity between TUBA1A and TUBA1B used in this study was 99.56%, and between TUBB3 and TUBB5, it was 92.57%.” (L451-L472).
- In Figure S2, highlight the differences among the sequences. Also show the secondary structural elements along with sequence.
(Answer)
I have added secondary structure information to the sequence in Figure S2 (now Figure S4), based on the structure of the modeling template.
- Mention the percentage of sequence similarity between the structures used for modeling in the modelling section.
(Answer)
The sequence similarity between the structures used for modeling in the modelling section were added to the modeling section as follows: “The sequence identity between the TUBA in 3RYC and the TUBA1A and TUBA1B used in this study was 99.56% and 100%, respectively. Additionally, the sequence identity between the TUBB in 3RYC and the TUBB3 and TUBB5 used in this study was 92.13% and 94.82%, respectively.” (L467-L470)
- Include information of any structural validation performed on the modelled structures.
(Answer)
Following your comments, we used MolProbity to validate our homology modeling-derived structures, which comprehensively evaluates 3D structural quality, considering aspects like clashes and conformational validity of main and side chains. The obtained structure exhibited a lower MolProbity Score than the template structure, indicating superior quality despite the template's 2.1 Å resolution crystal structure. Also, the core regions of the two modeled structures (TUBA:1-436, TUBB:1-427) showed a remarkably low Ca-RMSD of 0.344Å, and both interface loops were in StateA. Thus, the mobility differences in MD simulations are attributed to dynamics variances, not initial structure artifacts. These are added in the modelling section as follows: “The structures of TUBB3-TUBA1A and TUBB5-TUBA1B, obtained through homology modeling, underwent quality assessment using MolProbity [33]. Each structure received a MolProbity Score of 1.86 and 1.94, respectively, which are lower than the score of 2.02 for the template structure 3RYC. A lower MolProbity score indicates higher structural quality. Consequently, both structures generated through modeling exhibited superior quality compared to the crystal structure employed as the primary template. The Cα-RMSD values between modeling structures is 0.344Å. In Figure S3, we depicted the modeling structures and the template structure from PDB_ID 3RYC. Furthermore, in both modeling structures, the T7-loop of TUBB was in State A, as depicted in Figure 6. From these results, it is evident that the homology modeling performed in this study successfully yielded high-quality and highly similar structures of TUBB3-TUBA1A and TUBB5-TUBA1B.” (L482-L493)
- Calculate RMSF to understand the individual domain atomic fluctuations.
(Answer)
I have added a Cα-RMSF figure to Figure S7. Comparing this figure with the sequence and secondary structure location in Figure S4 supports the Cα-RMSD result that the secondary structure region is stable, and I have added that to the text. (L147-L149)
- Provide snapshots from MD simulations by highlighting the differences between initial structure and final structures. Figure 2 RMSD plots show high domain movements in some of the simulations.
(Answer)
In Figure 2, we have added overlays of the initial structure with three structures from each MD that have smaller RMSDs with respect to the final structure. Additionally, we have included overlays of structures at the positions with large RMSD observed in Figure 2a and b. Furthermore, the discussion of these figures is added to the text (L151-L155).
- Figure 4 and Figure 6, What does the y-axis label “Flames” means. Do the authors intend to write “Frames”.
(Answer)
We have corrected "Flame" to "Frame".
- Figure 6, the authors represent State A and State B, and their preference for different dimers. Is this order observed in all the 5 sets of simulations? If not, what is their probability?
- Also, represent the transformation of State A to State B during the simulations and discuss it in the manuscript. Time series plots of the atomic distances between the gln11 to gln245, gln245 to val353 and gln11 to val353 would explain this.
(Answer)
We have added a figure plotting the distances of CD@Gln245(TUBB)-CD@Gln11(TUBA) (d1) and CD@Gln245(TUBB)-CA@Gln353(TUBB) (d2) for all MD simulations conducted (Figure S12). In the figures, the interatomic distances d1 and d2 are represented by magenta and cyan lines. In these figures, the graph plot and State A and B have the following relationship: State A: d1 < d2, State B: d2 < d1. Moreover, we defined states based on the distances derived from Figure S12, where a state with CD@Gln245(TUBB)-CD@Gln11(TUBA) below 5 angstroms is termed State A and a state with CD@Gln245(TUBB)-CA@Gln353(TUBB) below 5 angstroms is termed State B. We calculated the proportion of frames in each state. All MD results contained both State A and State B, and notably, in TUBB3-TUBA1A, simulations with a majority of State A were prevalent, while in TUBB5-TUBA1B, simulations predominantly containing State B were more common. However, it was also observed that there were simulations in TUBB3-TUBA1A where State B was major and vice versa in TUBB5-TUBA1B.
- In Figure 5, show the nature of the contact area in terms of positive, negative and hydrophobic etc. This will provide a more comprehensive understanding of the PPI.
(Answer)
Following the reviewer's instructions, we have added Figure S8, which categorizes and color-codes the amino acids at the interface shown in Figure 5 into five types: positive charge, negative charge, polar, and hydrophobic.
From this figure, it is evident that, at the interface of TUBA and TUBB in the two tubulin dimers, the surface shape and the positions of charged and polar amino acids are in a complementary relationship with each other. Moreover, when comparing TUBA1A with TUBA1B and TUBB3 with TUBB5 across the two tubulin dimers, it is observed that there are no discernible differences in the distribution of amino acid types. These discussions have been added to the Discussion section as followings: “Additionally, as depicted in Figure S8, amino acid mutations at the interfaces of the aforementioned neuronal and ubiquitous tubulin dimers do not induce alterations in the distribution of amino acid types (positive, negative, polar, and hydrophobic). Consequently, it is improbable that the displacement of these amino acids would di-rectly instigate a modification in the interface contacts of both dimers.” (L295-L299).
- Improve the English writing in most of the sections.
(Answer)
We have worked to simplify each sentence/paragraph and eliminate redundancy to enhance the clarity and readability of the manuscript in English.
Reviewer 3 Report
The manuscript “Comparison of the molecular motility of tubulin dimeric isoforms: molecular dynamics simulations and diffracted X-ray tracking study” by Tsutomu Yamane et al. investigated differences in the molecular motility of tubulin dimers composed of various gene isoforms using diffracted X-ray Tracking analysis. Results revealed that the neuronal dimer (TUBB3-TUBA1A) had lower molecular motility around the vertical axis compared to the ubiquitous dimer (TUBB5-TUBA1B), likely due to changes in interface contacts identified by molecular dynamics (MD) simulations. The study highlights a novel phenomenon in the distinct pico-meter-scale molecular motility of neuronal and ubiquitous tubulin dimers. The study is robust, and the presentation is very good. I think this well-organized manuscript is worthy of publication in Int. J. Mol. Sci. I only have some concerns for the authors to address before I can endorse its publication.
1. I’m afraid that RMSDs (Figs. 2 & S4) are inadequate to describe dimer stability. RMSDs lack the sensitivity to detect significant changes at dimer interfaces. Interfaces may be immobilized even with plateaued dimer/monomer RMSDs, particularly in Cα RMSDs. It is more meaningful to assess the dimer interface by calculate its area or counting interface contacts versus time.
2. PCA analyses are valuable in this study, yet I have few concerns:
1) The authors stated that “the PCA plot of TUBB5-TUBA1B (Figure 3a) was wider and its occupancy was lower than that of TUBB3-TUBA1A (Figure 3b). This suggests that TUBB5-TUBA1B is more flexible than TUBB3-TUBA1A in terms of fluctuations in the relative positions of TUBA and TUBB” (page 5, lines 159–162). There are typos as the reference to figure panels are wrong. The PCA of TUBB5-TUBA1B should be “Figure 3b” instead of “Figure 3a”.
2) Comparing conformational plasticity through PCA makes sense if the MD trajectory of system B is projected onto the PCs from the MD trajectory of system A. From the manuscript it’s unclear if Figure 3a & Figure 3b were plotted against the same PCs. Please clarify, and if not, redo the analysis. For homologous systems, perform PCA analysis on aligned sequences.
3. About interface contacts, were those contact frequencies plotted in Figures S5 and S6 averaged from 5 independent runs? If yes, please report standard errors. This information is essential for assessing if the stability of a contact differs significantly between the two systems, e.g., Gln245–Asn246 in Figure S5a.
4. The Materials and Methods section needs some clarifications and additional references.
1) Cite [W. L. Jorgensen, et al. J. Chem. Phys. 1983, 79 (2), 926–935] for the Jorgensen’s original TIP3P water model.
2) Why did the authors use the V-rescale thermostat for temperature control? The Amber99SBnmr1-ildn force field was developed with the Nosé–Hoover thermostat. Note that one should use the same MD settings, including thermostat, barostat, non-bonded interactions, etc., as those employed in the force field development to guarantee reliable conformational sampling.
3) How did the authors calculate Coulombic interactions? Was the Particle Mesh Ewald used?
4) How did the authors calculate van der Waals (vdW) interactions? Did the authors potential- or force-switch the vdW? What was the switch range?
5) What was the step length? Was the LINCS algorithm employed?
Author Response
Reviewer 3
Thank you for your detailed peer review. As instructed, we have added more detailed information on our MD simulation methodology. In response to your request in comment 1, we also provide a plot of the number of contacts at the tubulin dimer interface as a function of time in the MD simulation, as we did in this study. We have also responded to the other comments we received below.
- I’m afraid that RMSDs (Figs. 2 & S4) are inadequate to describe dimer stability. RMSDs lack the sensitivity to detect significant changes at dimer interfaces. Interfaces may be immobilized even with plateaued dimer/monomer RMSDs, particularly in Cα RMSDs. It is more meaningful to assess the dimer interface by calculate its area or counting interface contacts versus time.
(Answer)
In this paper, we intended to convey two main points through the Cα-RMSD.
Firstly, we wanted to demonstrate that the structures of each α-tubulin and β-tubulin monomer that constitute the two tubulin dimers used in this study are stable. This is suggested by the sufficiently small Cα-RMSD of each tubulin dimer shown in Figure S4 (now Figure S6). Additionally, in the current revision, a figure of Cα-RMSF has been added as Figure S7, which also implies the structural stability of the core regions containing secondary structures.
Secondly, we aimed to show that there is a change in the relative positions between α-tubulin and β-tubulin in the two tubulin dimers used in this research. This is indicated by the magnitude of β-tubulin’s Cα-RMSD when fitted to α-tubulin in Figure 2. Following your guidance, we also present a figure showing the temporal change of the number of contacts at the interface.
As shown in the figure, the number of interface contacts for both TUBB3-TUBA1A and TUBB5-TUBA1B fluctuates within a range of approximately 200 to 500 over time. Such changes in the contacts at the interface suggest the presence of relative motion between the two monomers of the tubulin dimer. Moreover, in this study, Figure 2 demonstrates not only the relative positional change between monomers of the two tubulin dimers but also discusses the difference in mobility between the two tubulin dimers, TUBB3-TUBA1A and TUBB5-TUBA1B, in the PCA analysis shown in Figure 3.
We are not claiming the stability of the tubulin dimer from the RMSD in our paper, as per your comment. Likely, the misunderstanding arose from the statement, "The RMSD of Ca atoms in core regions of tubulin molecules distributed between about 1 and 2 Å suggesting both dimers' tubulin core structures were stable during the MD simulations (Figure S4)." (L146-L148). We have revised the description of Cα-RMSD, including this section, to address the misinterpretation. We revised as follows: “The RMSD values for the Cα atoms situated in the core region of each tubulin exhibited a range between 1 to 2 Å, signifying the stability of the core region (Figure S6). This stability was further underscored by the Cα-RMSF values from the simulations (Figure S7). The Cα-RMSF analysis showed flexibility primarily in the loop regions, exceeding 2 Å, while the rest of the structure fluctuates under 2 Å.” (L145-L149)
- PCA analyses are valuable in this study, yet I have few concerns:
1) The authors stated that “the PCA plot of TUBB5-TUBA1B (Figure 3a) was wider and its occupancy was lower than that of TUBB3-TUBA1A (Figure 3b). This suggests that TUBB5-TUBA1B is more flexible than TUBB3-TUBA1A in terms of fluctuations in the relative positions of TUBA and TUBB” (page 5, lines 159–162). There are typos as the reference to figure panels are wrong. The PCA of TUBB5-TUBA1B should be “Figure 3b” instead of “Figure 3a”.
(Answer)
Following the instructions, we have corrected the typos and modified the manuscript by adding a more detailed explanation of the figures as follows. “In the PCA plot of TUBB3-TUBA1A (Figure 3a), a unimodal distribution was observed, characterized by a peak situated near (0,0) and exhibiting a presence frequency of 3.0%. Conversely, TUBB5-TUBA1B (Figure 3b) displayed a distribution with a flattened peak and a presence frequency of less than 1.5%. These observations suggest that TUBB5-TUBA1B is inclined to adopt a more diverse states concerning the relative posi-tions of alpha- and beta-tubulin compared to TUBB3-TUBA1A.” (L171 – L177).
2) Comparing conformational plasticity through PCA makes sense if the MD trajectory of system B is projected onto the PCs from the MD trajectory of system A. From the manuscript it’s unclear if Figure 3a & Figure 3b were plotted against the same PCs. Please clarify, and if not, redo the analysis. For homologous systems, perform PCA analysis on aligned sequences.
(Answer)
In Figure 3a and 3b, which illustrate the principal component analysis (PCA) conducted in this study, all trajectories obtained from each of the five runs of MD simulations for TUBB3-TUBA1A and TUBB5-TUBA1B were plotted on the plane defined by the principal axes, PC1 and PC2. Therefore, Figures 3a and 3b share the same PC axes. We have added details regarding the determination of the principal axes to the legend of Figure 3. “The principal axes in these figures are those obtained from the trajectories of all MD simulations in TUBB3-TUBA1A and TUBB5-TUBA1B.” (L184-185)
- About interface contacts, were those contact frequencies plotted in Figures S5 and S6 averaged from 5 independent runs? If yes, please report standard errors. This information is essential for assessing if the stability of a contact differs significantly between the two systems, e.g., Gln245–Asn246 in Figure S5a.
(Answer)
Following your instructions, we have appended the standard deviation of the contact frequency for each MD run to the legend in Figure S5 (now Figure S9) and Figure S6 (now Figure S10).
- The Materials and Methods section needs some clarifications and additional references.
1) Cite [W. L. Jorgensen, et al. J. Chem. Phys. 1983, 79 (2), 926–935] for the Jorgensen’s original TIP3P water model.
(Answer)
Following your instructions, I have cited the TIP3P water model paper by Jorgensen, et al.
2) Why did the authors use the V-rescale thermostat for temperature control? The Amber99SBnmr1-ildn force field was developed with the Nosé–Hoover thermostat. Note that one should use the same MD settings, including thermostat, barostat, non-bonded interactions, etc., as those employed in the force field development to guarantee reliable conformational sampling.
(Answer)
Thank you for your comment. We have no clear reason to use the V-rescale method here.
3) How did the authors calculate Coulombic interactions? Was the Particle Mesh Ewald used?
(Answer)
The Coulomb interaction utilized Partivle mesh Ewald. This has been added in Material and Methods.
4) How did the authors calculate van der Waals (vdW) interactions? Did the authors potential- or force-switch the vdW? What was the switch range?
(Answer)
The calculation of the van der Waals (vdw) interaction utilized a short-range van der Waals cutoff of 1.0 nm. This information has been added to the Materials and Methods section.
No potential-switch or force-switch was used.
5) What was the step length? Was the LINCS algorithm employed?
(Answer)
The step length is 0.002 ps and the LINCS algorithm was used. This has been added in the Material and Methods section.
Round 2
Reviewer 1 Report
Regarding my comments, the authors have adequately revised the manuscript.
Reviewer 2 Report
The authors answered all the questions raised and incorporated the modifications at relevant sections. It can be accepted in the present form.
Reviewer 3 Report
In the revised manuscript, the authors have addressed all my questions and concerns. I am satisfied with the content now. It is recommended to publish this manuscript in its current form.